# Methodological and Biological Factors Influencing Global DNA Methylation Results Measured by LINE-1 Pyrosequencing Assay in Colorectal Tissue and Liquid Biopsy Samples

**DOI:** 10.3390/ijms231911608

**Published:** 2022-10-01

**Authors:** Krisztina A Szigeti, Barbara K Barták, Zsófia B Nagy, Sára Zsigrai, Márton Papp, Eszter Márkus, Peter Igaz, István Takács, Béla Molnár, Alexandra Kalmár

**Affiliations:** 1Department of Internal Medicine and Oncology, Faculty of Medicine, Semmelweis University, 1083 Budapest, Hungary; 2Centre for Bioinformatics, University of Veterinary Medicine Budapest, 1078 Budapest, Hungary; 3Department of Anaesthesia and Intensive Care, Pest County Flor Ferenc Hospital, 2143 Kistarcsa, Hungary; 4MTA-SE Molecular Medicine Research Group, Eötvös Loránd Research Network, 1083 Budapest, Hungary; 5Department of Endocrinology, Faculty of Medicine, Semmelweis University, 1083 Budapest, Hungary

**Keywords:** global DNA methylation, long interspersed nuclear element-1, LINE-1 bisulfite pyrosequencing, colorectal cancer

## Abstract

Long interspersed nuclear element 1 (LINE-1) bisulfite pyrosequencing is a widely used technique for genome-wide methylation analyses. We aimed to investigate the effects of experimental and biological factors on its results to improve the comparability. LINE-1 bisulfite pyrosequencing was performed on colorectal tissue (*n* = 222), buffy coat (*n* = 39), and plasma samples (*n* = 9) of healthy individuals and patients with colorectal tumors. Significantly altered methylation was observed between investigated LINE-1 CpG positions of non-tumorous tissues (*p* ≤ 0.01). Formalin-fixed, paraffin-embedded biopsies (73.0 ± 5.3%) resulted in lower methylation than fresh frozen samples (76.1 ± 2.8%) (*p* ≤ 0.01). DNA specimens after long-term storage showed higher methylation levels (+3.2%, *p* ≤ 0.01). In blood collection tubes with preservatives, cfDNA and buffy coat methylation significantly changed compared to K3EDTA tubes (*p* ≤ 0.05). Lower methylation was detected in older (>40 years, 76.8 ± 1.7%) vs. younger (78.1 ± 1.0%) female patients (*p* ≤ 0.05), and also in adenomatous tissues with *MTHFR* 677CT, or 1298AC mutations vs. wild-type (*p* ≤ 0.05) comparisons. Based on our findings, it is highly recommended to consider the application of standard DNA samples in the case of a possible clinical screening approach, as well as in experimental research studies.

## 1. Introduction

Global DNA hypomethylation is an early molecular event in colorectal cancer (CRC) leading to genetic instability through the reactivation of mobile genetic elements, especially of the long interspersed nuclear element 1 (LINE-1) sequences [1,2]. Global DNA and LINE-1 methylation levels have a prognostic value [3,4,5,6,7]; moreover, LINE-1 methylation also possesses diagnostic potential in a cancer screening procedure analyzed in plasma samples obtained from patients with colorectal tumors [8,9,10]. Hence, it is essential to measure LINE-1 methylation levels in experimental and clinical fields in a reliable, comparable, and repeatable way. The transposable LINE-1 sequences are approximately 6 kilobase pairs long and contain two open-reading frames (ORFP1, ORFP2) [11], which are transcriptionally repressed by genome-wide DNA methylation [12,13,14]. 

Different biological factors such as age [15], sex [16], or regular physical activity [17,18,19] can affect global DNA methylation values, hence also the results of the applied estimating methods. Furthermore, methylenetetrahydrofolate reductase (MTHFR) also has an important role, as it is a key enzyme of one-carbon metabolism, in which S-adenosylmethionine, the main methyl-donor molecule for DNA methylation, is produced [20]. Therefore, the status of the two most common MTHFR gene mutations (C677T [21] and A1298C [22]) has a substantial impact on DNA methylation [23,24]. These above-mentioned biological aspects were chosen to be analyzed in relation to their influence on LINE-1 methylation results.

The gold standard method for global DNA methylation estimation is liquid chromatography combined with tandem mass spectrometry (LC-MS/MS), with which 5-methylcytosine in proportion to all cytosine residues can be detected [25]. However, the needed expertise and equipment are not widely available. Enzyme-linked immunosorbent assay (ELISA) specific to 5-methylcytosine is a quick and simple, widely used method for overall DNA methylation estimation, but it is able to measure only rough differences in DNA methylation; moreover, it shows high inter- and intraplate variations [25]. In parallel, LINE-1 methylation values correlate well with the LC-MS/MS results [25,26,27,28]; therefore, several protocols have been introduced to assess the methylation level of LINE-1 sequences. Global DNA methylation values estimated on the basis of LINE-1 bisulfite pyrosequencing can vary in a certain range [29], but it showed the lowest variability and the highest signal-to-noise ratio compared to other global DNA methylation analyzing methods such as combined bisulfite restriction analysis, MethyLight, or methylation-specific PCR techniques [30]. Furthermore, LINE-1 pyrosequencing is one of the most cost-effective, reliable techniques that needs a relatively low amount of bisulfite-converted input DNA [25,30]. Therefore, it is a widely applied method for global DNA methylation evaluation [31,32,33,34,35,36,37]. Taken together, these literature data led us to prefer pyrosequencing in our previous works, resulting in the realization of the importance to investigate methodological and biological circumstances that may impact the experimental results of LINE-1 pyrosequencing. This method relies on LINE-1-specific primers used for pyrosequencing and the antecedent bisulfite-specific PCR of a region in the LINE-1 promoter containing three CpG dinucleotides following bisulfite conversion. Pyrosequencing of the amplicons reveals the methylation level of the three CpG sites calculated based on the proportion of detected cytosines and thymines in the given positions [38]. 

Methodological factors such as different manners of sample collection and storage conditions can vary during an experimental or clinical investigation, which may have a significant impact on the methylation results. Since global DNA methylation analysis results can vary in a certain interval [29], but alterations lower than 5% in LINE-1 methylation can hold monitoring potential in post-operative CRC patients [8], standardized circumstances are needed in a possible diagnostic application of global DNA methylation estimation by LINE-1 pyrosequencing. The accuracy of this technique has already been investigated in plasma, colorectal FFPE tissue, and buffy coat samples [29,30,34]. However, a comprehensive analysis of the possible technical, experimental, and biological effects on LINE-1 methylation can provide useful information about the critical points of this procedure. Moreover, even though the application of unmethylated and methylated standards is not involved in the applied protocol, they may improve the comparability of the LINE-1 methylation values measured in different batches. 

In this study, we aimed to shed light on the importance of the methodological, technical, and biological details which may affect the comparability of LINE-1 bisulfite pyrosequencing results in colorectal tissue, buffy coat, and cell-free DNA (cfDNA) samples. 

Firstly, we aimed to investigate the methylation values of distinct CpG sites in the LINE-1 promoter region analyzed by the Pyromark Q24 CpG LINE-1 kit (Qiagen, Hilden, Germany). Moreover, our goal was to examine the technical factors such as different sample collection techniques by collecting endoscopic (E) and surgical (S) colorectal tissue biopsies, along with distinct fixation methods such as formalin-fixed paraffin-embedded (FFPE), and matched fresh frozen tissue (FF) specimens. Furthermore, long-term and short-term storage of DNA samples isolated from tissue biopsies and the influence of whole blood storage time and temperature until blood separation were also analyzed in buffy coat and plasma samples taken into standard K3EDTA and blood collection tubes with cfDNA preservation agents. The Cell-Free DNA Collection Tube (Roche, Mannheim, Germany) was developed for cfDNA stabilization and peripheral blood cell lysis inhibition. Preservation agents can cause covalent modifications of the DNA, leading to methylation alterations [39,40]. Hence, we aimed to examine the possible influence of Roche collection tube application on LINE-1 methylation compared to the standard K3EDTA blood collection tube. Finally, the impact of biological factors such as aging, sex, MTHFR status, and physical activity was also analyzed in our study. 

## 2. Results

### 2.1. Methylation Values of Different LINE-1 CpG Positions 

In colon tissue specimens, significantly elevated methylation values were detected in CpG position 1 (CpG 1) of non-tumorous samples (N, AD-NAT, CRC-NAT, and IBD) compared to CpG position 2 (CpG 2), 3 (CpG 3), and mean methylation values (*p* ≤ 0.01) (Figure 1A, left). However, no such significant alterations were observed within colon adenoma and carcinoma groups (AD and CRC) (Figure 1A, right). As a control sample group, paired buffy coat specimens were also analyzed, and significant methylation differences were noticed in the comparison of CpG 1 vs. 2, 3, and mean methylation values when assessing healthy individuals (Figure 1B, left) (*p* ≤ 0.01), as well as patients with colorectal tumors (Figure 1B, right) (*p* ≤ 0.01).

### 2.2. The Influence of Different Tissue Fixation and Sample Collection Methods

Investigations about measuring the possible influence of different fixation methods and sample collection techniques were performed on tumorous (AD and CRC) and non-tumorous (N, AD-NAT, CRC-NAT) tissue samples (Figure 2A). A significant LINE-1 methylation decrease was detected in FFPE samples (73.0 ± 5.3%) compared to paired FF specimens (76.1 ± 2.8%) (*p* ≤ 0.01) (Figure 2A). A slightly lower LINE-1 methylation was observed in samples collected during surgery (S, 71.4 ± 6.5%) compared to endoscopic specimens (E, 72.1 ± 6.9%) (Figure 2B).

### 2.3. The Impact of Different Storage Conditions on LINE-1 Methylation 

Differences between average methylation were calculated and represented in the case of matched samples in the following comparisons. 

In colorectal tissue samples, LINE-1 methylation alteration of −0.9% was observed following short-term −20 °C storage of the isolated DNA, while a significantly increased LINE-1 methylation level (+3.2%) was detected in the case of long-term storage (*p* ≤ 0.01) (Figure 3A). 

In the comparison of Roche vs. K3EDTA blood collection tubes, methylation levels were significantly elevated (+2.1%) in cfDNA and significantly decreased (−0.4%) in buffy coat samples digested for 1 h (*p* ≤ 0.05) (Figure 3B). In 2 h ProtK buffy coat specimens, a non-significant, minor decrease was observed (−0.5%). 

Whole blood samples collected in K3EDTA tubes, and stored at room temperature, showed no significant alteration in buffy coat LINE-1 methylation level after 3 (+0.08%) and 6 (−0.6%) hours. At 4 °C, no changes could be detected (−0.2%, −0.1%) over time. In the case of Roche collection tubes, after 1 h ProtK digestion of buffy coat fractions, −0.2% and −1% methylation alterations were observed after 3 h, and 6 h, respectively. Following 2 h of ProtK digestion, LINE-1 methylation changes of +0.8% and +1.2% were found over time (Figure 3C). 

CfDNA LINE-1 methylation showed a non-significant decline in the case of K3EDTA collection tubes (RT 3 h: −0.1%, 6 h: −1%; 4 °C 3 h: −1.6%, 6 h: −1%), while in Roche collection tubes a moderate elevation of +1.4% and a minor decrease of −0.7% were observed after 3 h and 6 h storage at room temperature, respectively (Figure 3D). 

### 2.4. Methylation Values of Unmethylated and Methylated Standards

Methylation values of bisulfite-converted unmethylated and methylated control DNA samples showed no remarkable alterations in repeated measurement after two years compared to the base experiment (Figure 4). Higher standard deviations were observed in the case of methylated controls compared to the unmethylated ones; however, there was no significant difference between the variances in the batches measured after two years compared to the first experiments according to an F-test (Figure 4). 

### 2.5. Tissue LINE-1 Methylation Alterations Related to Aging, Sex, and Physical Activity 

Investigation of LINE-1 methylation age dependence revealed no significant correlation in healthy colorectal tissue biopsies (⍴ = −0.18, *p* = 0.25) (Figure 5A). Significantly decreased LINE-1 methylation was detected in the samples of female patients over 40 years old (76.8 ± 1.7%) compared to those under the age of 40 (78.1 ± 1.0%) (*p* ≤ 0.05) (Figure 5B). The recently performed physical activity caused no significant alteration in LINE-1 methylation in cfDNA; however, a moderate decrease could be observed during (79.7 ± 3.4%, mean heart rate: 134.5 beats per minute) and following (79.4 ± 2.4%, mean heart rate: 91.2 beats per minute) the physical activity compared to the rest phase (80.7 ± 2.3%, mean heart rate: 76.4 beats per minute) (Figure 5C). 

### 2.6. LINE-1 Methylation Differences in MTHFR Mutations

The influence of MTHFR heterozygous mutations (HE) showed no significant alteration in the case of healthy colorectal tissue samples (Figure 6A), while significantly lower methylation levels were detected in AD HE specimens of both mutant alleles (C677T, A1298C) compared to the wild-types (WTs) (*p* ≤ 0.05) (Figure 6B). A moderate LINE-1 methylation elevation in CRC samples could be found in heterozygous C677T and A1298C specimens compared to WTs (Figure 6C). 

## 3. Discussion

Global DNA hypomethylation is a molecular driving force in cancer development through mobile genetic elements – such as LINE-1 retrotransposons - reactivation [1,2,41]; hence, LINE-1 methylation level possesses important information as a prognostic and diagnostic value in cancer diseases, including colorectal cancer [8,9,10,42]. In this study, our aim was to comprehensively examine the potential methodological, technical, and biological factors that can influence the LINE-1 methylation measurements in a colon tissue, buffy coat, and plasma sample cohort. 

First, we focused on the different technical factors of our study. Intriguing findings were observed regarding the LINE-1 methylation levels of the three individual CpG sites involved in the Pyromark Q24 CpG LINE-1 assay (Qiagen). In CpG 1 position (CpG 328), a significantly higher 5-methylcytosine level was observed in non-tumorous samples compared to the CpG 2 (CpG 321), 3 (CpG 318), and mean methylation values. Furthermore, the highest methylation difference was observed in the case of CpG 1 during the LINE-1 hypomethylation process in tumorous samples. As CpG sites of the LINE-1 promoter region have different demethylation tendencies [43] and are characterized by distinct methylation profiles in diverse cancer types [44,45], we assume a possible functional role of CpG 1 in colorectal cancer. In accordance with these results, it is worth paying special attention to the methylation alteration of this particular CpG site in a comparative investigation. In our further analyses, the investigated conditions had no significant effect specifically on CpG 1 methylation; hence, mean methylation percentages were discussed. 

Our analysis regarding the effect of the chosen fixation method revealed significantly decreased methylation levels in FFPE compared to paired FF samples. According to the recent literature, FFPE specimens have elevated methylation levels due to the unmethylated cytosines which are not bisulfite-convertible because of the formalin fixation [45,46]. However, unmethylated and methylated cytosines can undergo deamination during formalin fixation, resulting in uracils or thymines, respectively [47], leading to decreased DNA methylation levels, which can explain our results. 

In the investigation of disparate sample collection methods, surgical and endoscopic samples were analyzed. A non-significantly lower range of LINE-1 methylation results was found in surgical samples compared to endoscopic ones. This observation could be related to the different cell compositions of the specimens collected by distinct methods, as colonoscopic samples are superficial, containing proportionately more epithelial cells than surgically collected tissue. Mesenchymal and muscle cells possess lower global DNA methylation levels than colon tissue epithelium [48].

Comparative studies commonly work with long-term stored specimens, while routine clinical analyses are usually performed on freshly collected samples; therefore, we investigated both conditions. Firstly, we aimed to evaluate the LINE-1 methylation levels of DNA standards, along with DNA samples isolated from the same biological tissue biopsies and kept at −20 °C for two days or two years. In the present study, short-term storage (two days) resulted in no significant LINE-1 methylation alteration, in agreement with the study of Irahara et al. [29]. However, according to their data, it is recommended to conduct repeated measurements to estimate the exact LINE-1 methylation values more precisely [29]. Measurements performed after two years of −20 °C storage revealed significant methylation elevation in our experiments. Contradictory results were noticed by Gosselt et al. using liquid chromatography coupled with tandem mass spectrometry, according to which global DNA methylation reduction was detected in DNA specimens isolated from blood samples after 18 months of storage at −20 °C [49]. Our analyses of long-term storage of bisulfite-converted, unmethylated, and methylated DNA standards revealed no significant impact on LINE-1 methylation level. Hence, besides the storage time, we assume that the usage of bisulfite conversion kits purchased in distinct production cycles may also lead to dissimilarities in the values of LINE-1 methylation. Therefore, the involvement of unconverted unmethylated and methylated DNA controls could be an advantageous opportunity to monitor the whole procedure and assess the batch effect throughout the applied project.

In the case of freshly collected whole blood samples, the impacts of standard K3EDTA and blood collection tubes with stabilization reagents (Roche collection tubes) were investigated in buffy coat and plasma samples, along with the influence of different storage conditions. In the case of Roche collection tubes, significantly higher methylation levels were observed in cfDNA compared to standard K3EDTA tubes; moreover, significantly lower methylation values were found in buffy coat samples in the Roche vs. K3EDTA comparison analyzed by our research group. Accordingly, our previous study revealed significantly altered DNA methylation patterns of cfDNA samples obtained from CRC patients in stabilization blood collection tubes (Streck) compared to the conventional K3EDTA [50]. However, no significant methylation alteration was detected in white blood cells in the presence of DNA preservation agents, published by Bulla et al. [51]. 

Distinct storage temperatures (RT and 4 °C) and disparate storage times (0, 3, and 6 h) were examined as different storage variables in buffy coat and cfDNA specimens collected in conventional K3EDTA and Cell-Free DNA Collection Tubes (Roche). No significant LINE-1 methylation alterations could be found either in buffy coat or cfDNA samples according to our investigations. However, published by Shiwa et al., examination of buffy coat specimens conducted by Illumina Infinium Human Methylation 450 array revealed a biased genome-wide methylation pattern as a consequence of 4 °C storage attributed to altered cell composition [52]. Moreover, storage time was also a considerable factor due to the significant global DNA methylation decline in white blood cells after 3 days of RT or 4 °C storage of blood samples based on the data observed by Huang et al. [53]. In contrast with our results about cfDNA LINE-1 methylation, 72 h of storage of whole blood samples revealed slightly increased overall DNA methylation of cfDNA in EDTA tubes using reduced representation bisulfite sequencing according to a recent study [54]. Therefore, we would like to emphasize that in an intraindividual monitoring procedure, distinct preanalytical conditions can lead to remarkable LINE-1 methylation changes; consequently, standardization of these critical points of sample preparation is crucial. 

In the second part of our study, we aimed to analyze the influence of distinct biological variables such as age, sex, physical training, and MTHFR status on the LINE-1 methylation pattern. No significant correlation was found in normal colon tissue samples between age and LINE-1 methylation according to our data; however, a significant LINE-1 methylation decrease could be detected in the case of female patients over 40 years old compared to those under 40. In a previous study, significantly lower LINE-1 methylation of the normal colonic mucosa was observed in the oldest quartile of the patients compared to the youngest; however, analysis of all quartiles revealed no significant differences, affirming our data [55]. Furthermore, no significant LINE-1 methylation differences were noticed between sexes in our experiments, confirming the results of a previous publication regarding colon tissue samples [56], while higher LINE-1 methylation values of blood samples were published, in males compared to females by El-Maarri et al. [57]. According to these findings, a linear clinical investigation is possibly not influenced by aging, but age and sex dependency should be taken into account in experimental research. 

Regular physical training leads to an elevated LINE-1 methylation level [17,19]; hence, we involved the analysis of the influence of physical activity on LINE-1 methylation level, since, in a clinical application, it is highly important to know whether activities related to the daily routine could affect LINE-1 methylation results. A non-significant, minor LINE-1 methylation decrease was found during and following recent physical activity compared to the resting phase in our investigations. The biases were not significant; however, they may modify the methylation results in clinical sampling, which highlights the importance of the standardization of physical activity as well in a possible clinical routine investigation. 

Besides these biological aspects, it is worth considering the mutational status of the MTHFR gene (C677T and A1298C), since these highly frequent mutations lead to lowered enzyme activity and S-adenosylmethionine availability [23,24]. No remarkable alterations in healthy colorectal tissue were noticed in our investigation; meanwhile, higher LINE-1 methylation levels were detected in the case of samples with higher MTHFR activity in normal colonic mucosa according to Iacopetta et al. [55]. Our examinations revealed significantly lower LINE-1 methylation values in both heterozygous MTHFR alleles of adenomatous samples compared to wild-type specimens. In addition, moderately higher methylation levels were detected in 677CT and also in 1298AC mutant colorectal cancer tissue biopsies in comparison with wild-type samples. These results are affirmed by the findings of Sohn et al., according to which elevated DNA methylation was observed in the HCT116 colon cancer cell line with heterozygous C677T MTHFR alleles [58]. Therefore, in the case of a heterogeneous sample set with diverse MTHFR mutations, it is necessary to evaluate the results in light of the fact that MTHFR status can also affect the LINE-1 methylation level.

In conclusion, methodological and biological variables can impact the LINE-1 methylation pattern measured by pyrosequencing on a relevant level, which should be taken into account in clinical and research fields as well. 

In research, it is essential to collect and fix samples to be compared according to the same method; moreover, the experiments related to the same question are suggested to be conducted with kits purchased in the same production cycle. The importance of DNA standards applied from the beginning of the procedure is also highly emphasized, especially in a longitudinal examination.

In a medical investigation, standardization of the circumstances is crucial and more attention is needed, especially in the case of a multicenter sample collection. It is recommended to take the blood samples into the same type of collection tubes, store the samples according to the same protocol, and process them as soon as possible after the collection. Furthermore, the application of DNA standards is also suggested. 

More investigations should be carried out concerning the effect of physical activity to determine an optimal resting period before the blood draw if it has been proven necessary. In regard to the other analyzed biological factors, such as age, sex, and MTHFR status, they are not relevant in the comparison of samples obtained from the same patient during a medical investigation; however, it is highly recommended to take them into account in experimental research.

## 4. Materials and Methods

### 4.1. Sample Collection 

The investigated sample cohort was collected at the Department of Internal Medicine and Hematology, the 1 Department of Pathology and Experimental Cancer Research, Semmelweis University, and the Hungarian University of Sports Science, Budapest, Hungary. In this study, the following samples were involved: 180 fresh frozen (FF) and 25 formalin-fixed paraffin-embedded tissue (FFPE) biopsies, along with 38 and 12 blood specimens taken into standard K3EDTA Vacuettes (Greiner Bio-One Gmbh) and Cell-Free DNA Collection Tubes (Roche), respectively. The research has been approved by the local ethics committee (Regional and Institutional Committee of Science and Research Ethics; TUKEB Nr: 14383-2/2017/EKU). Written informed consent was obtained from all the involved patients prior to specimen collection. 

### 4.2. Tissue Samples

FF specimens consisted of 45 healthy (N), 23 normal adjacent tissue to colorectal adenoma (AD-NAT), 24 normal adjacent tissue to colorectal carcinoma (CRC-NAT), 37 colorectal adenomas (AD), 36 colorectal carcinomas (CRC), and 15 inflammatory bowel disease colorectal tissue (IBD) biopsies gathered during colonoscopy. 

In the case of 9 N, 12 AD, and 4 CRC, paired FFPE samples were also collected to examine the possible effects of formalin fixation and paraffin embedding on LINE-1 methylation.

To analyze the influence of sample collection methods on LINE-1 methylation levels, 36 CRC and 24 CRC-NAT samples were obtained during colonoscopy, along with 21 CRC and 21 CRC-NAT from the resected colon sections immediately after surgery. 

Furthermore, the impact of −20 °C storage of genomic DNA was also investigated on LINE-1 methylation level with repeated measurements performed on the same biological DNA specimens within two days (short-term, *n* = 27) and two years (long-term, *n* = 48). Moreover, bisulfite-converted unmethylated and methylated standard DNA samples (EpiTect Control DNA and Control DNA Set, Qiagen) were also involved in this examination.

Analyzed sample numbers are summarized in Table 1, while the clinicopathological and demographic data of the investigated patients are summarized in Appendix A.

### 4.3. Liquid Biopsy Samples

Blood samples were taken into standard K3EDTA blood collection tubes from 19 normal individuals, 5 athletes, and 10 AD and 10 CRC patients. 

The investigation of individual CpG sites’ methylation levels contained 15 N, 10 AD, and 10 CRC buffy coat samples as a control cohort to analyze the experienced phenomenon from colon tissue specimens in a different sample type. 

In the case of 4 healthy patients, liquid biopsies were also collected into Cell-Free DNA Collection Tubes (Roche) to investigate the possible impact of the preservation agent on the buffy coat and cfDNA LINE-1 methylation levels. Furthermore, to examine the effect of different storage times and temperatures, buffy coat and plasma fractions were separated immediately after the blood draw, after 3 and 6 h of storage at room temperature, and 4 °C in parallel. 

Moreover, whole blood samples were taken into K3EDTA tubes from 5 athletes, who performed a twenty-five minute-long run on a treadmill. The first blood samples were collected before the physical activity (rest), the second one at the highest heart rate (activity), and the last one at 30 min after the run, during the restitution period (restitution). 

The analyzed sample numbers are summarized in Table 1; furthermore, the clinicopathological and demographic data of the investigated patients are summarized in Appendix A.

The process of sample collection and the applied methods are presented in Figure 7.

### 4.4. DNA Isolation from Fresh Frozen, Formalin-Fixed Paraffin-Embedded Tissue, Buffy Coat, and Plasma Samples

Genomic DNA isolation was performed from fresh frozen and formalin-fixed, paraffin-embedded tissue biopsies using a High Pure PCR Template Preparation Kit (Roche) according to the manufacturer’s protocol. 

Buffy coat layers were separated from whole blood specimens by centrifugation at 1350 rcf for 12 min. Genomic DNA was isolated from 200 µL buffy coat fractions (High Pure PCR Template Preparation Kit, Roche) in line with the manufacturer’s instructions. In the case of the Cell-Free DNA Collection Tube (Roche), ProtK digestion was performed for one and two hours. 

Plasma fractions were separated from whole blood samples by two centrifugation steps (1350 rcf, 12 min). CfDNA isolation was carried out from 1–2 mL plasma samples with the Quick-cfDNA™ Serum & Plasma Kit (Zymo Research, Orange, FL, USA) according to the manufacturer’s recommendation. All DNA samples were stored at −20 °C until further examinations. 

A NanoDrop ND-1000 Spectrophotometer (Thermo Fisher Scientific, Orange, FL, USA) was applied to measure the concentration and purity (OD260/280, OD230/280) of both tissue and buffy coat genomic DNA, while quantification of cfDNA was carried out with a Qubit 1.0 fluorometer using a Qubit dsDNA HS Assay Kit (Thermo Fisher Scientific).

### 4.5. LINE-1 Bisulfite Pyrosequencing

Applying an EZ DNA Methylation-Direct Kit (Zymo Research), 500 ng DNA of each FF, 200 ng DNA of each FFPE tissue, and buffy coat sample, as well as approximately 20 ng cfDNA of each plasma specimen, were bisulfite-converted according to the manufacturer’s recommendation following DNA isolation. Using a Pyromark Q24 CpG LINE-1 Kit (Qiagen), a 146 base pair-long region of the LINE-1 promoter was amplified via bisulfite-specific PCR on Mastercycler EP Gradient S PCR machine (Eppendorf, Hamburg, Germany). Thermocycling protocol was the following: PCR activation at 90 °C for 15 min; 45 cycles of denaturation at 94 °C for 30 s, annealing at 50 °C for 30 s and 30 s extension at 72 °C; final extension lasting 10 min at 72 °C. The specificity of PCR was verified with electrophoresis using 2% agarose gel. Sample preparation and washing steps were carried out on a PyroMark Q24 Vacuum Workstation (Qiagen) followed by pyrosequencing on a Pyromark Q24 (Qiagen) instrument using a Pyromark Q24 CpG LINE-1 Kit (Qiagen) and PyroMark Gold Q24 Reagents (Qiagen). Methylation of three CpG dinucleotides (positions 318 (CpG position 3—CpG 3), 321 (CpG position 2—CpG 2), and 328 (CpG position 1—CpG 1) of LINE-1 sequence, GenBank accession number: X58075) was calculated as the percentage of cytosine nucleotides relative to the sum of cytosine and thymine nucleotides in a given position by Pyromark Q24 software v2.0.6. The methylation status was assessed individually at each CpG site and also as an average value of the three CpG positions in the case of colon non-tumorous and tumorous FF samples; meanwhile, only average methylation levels were presented in the investigation of other methodological and biological aspects. In those cases, detailed results about individual CpG positions’ methylation are summarized in Appendix A. The whole protocol was performed according to the manufacturer’s instructions.

### 4.6. Statistical Analyses 

Statistical significance (*p* ≤ 0.05) was determined by applying Prism8 software (GraphPad, San Diego, CA, USA). In multiple comparison tests, data with non-normal distribution were analyzed by Kruskal–Wallis and Dunn’s multiple comparison tests, while in the case of normal distribution, ANOVA followed by Brown Forsythe and Welch ANOVA tests was performed. In pairwise comparisons of unmatched sample groups, an unpaired *t*-test or Mann–Whitney test was performed depending on normal or non-normal data distribution. In the case of matched specimens, a paired *t*-test or Wilcoxon matched-pairs signed-rank test was conducted on parametric or non-parametric data, respectively. The correlation between age and LINE-1 methylation in colorectal tissue samples was calculated using a Spearman test.

## 5. Conclusions

Significant LINE-1 methylation changes were observed regarding different methodological aspects such as the different CpG sites in the LINE-1 promoter region, where significantly higher methylation values were detected in the case of the CpG 1 position (CpG 328) compared to CpG 2 (CpG 321), 3 (CpG 318), and mean methylations in non-tumorous samples, while in tumorous specimens no such differences could be observed. Regarding the fixation methods, significantly higher LINE-1 methylation levels were found in FF samples compared to FFPE. Furthermore, long-term storage at −20 °C of DNA samples isolated from tissue biopsies showed significant elevation. The application of blood collection tubes with preservation agents resulted in significantly increased methylation in cfDNA and significantly reduced methylation in buffy coat samples compared to the standard K3EDTA tubes. Moreover, age in female patients and MTHFR heterozygous alleles in AD samples also significantly impacted LINE-1 methylation results. Since LINE-1 bisulfite sequencing is a cost-effective, favorable, and widely used method for LINE-1 methylation examinations in research, as well as holds a possible diagnostic potential for colorectal cancer monitoring, it is important to pay special attention to standardizing the above-mentioned circumstances. Furthermore, we conclude that it is necessary to consider the application of internal DNA standards for clinical screening and for research purposes as well.

## Figures and Tables

**Figure 1 ijms-23-11608-f001:**
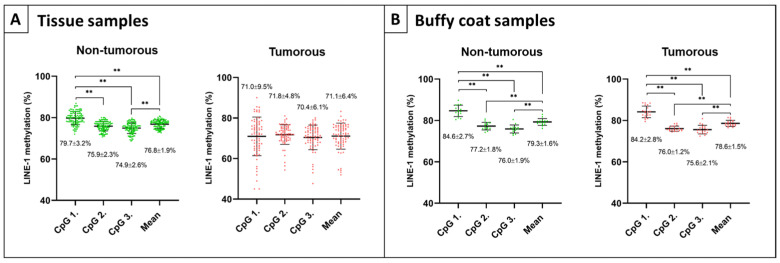
The methylation level of CpG sites of LINE-1 retrotransposon measured by the Pyromark Q24 CpG LINE-1 assay (Qiagen). (**A**) Significantly higher methylation level was observed in the case of the first CpG site compared to the second, third positions, and the mean methylation value in non-tumorous (N, AD-NAT, CRC-NAT, and IBD) tissue biopsies (** *p* ≤ 0.01) (**left**). Similar methylation levels of CpG positions were detected in AD and CRC tissue samples (**right**). (**B**) In buffy coat fractions of normal controls (**left**), along with patients with colorectal tumors (**right**), significantly elevated methylation value was detected in CpG 1 compared to positions 2, 3, and mean methylation (** *p* ≤ 0.01). Non-tumorous samples are represented in green, while tumorous specimens are marked in red.

**Figure 2 ijms-23-11608-f002:**
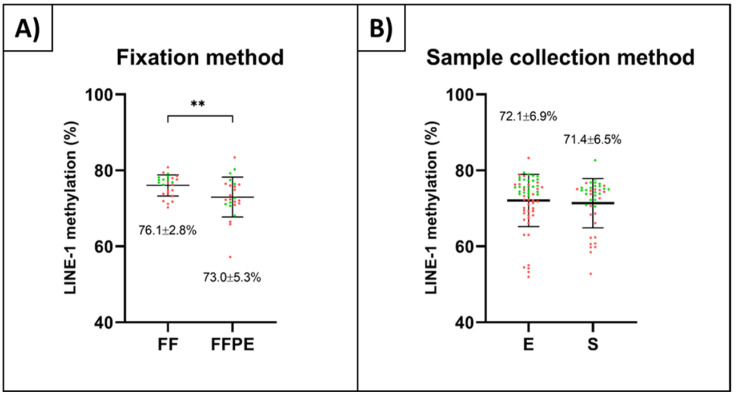
Influence of fixation methods and sample collection methods on LINE-1 methylation level. (**A**) FF samples (76.1 ± 2.8%) showed significantly higher LINE-1 methylation compared to FFPE specimens (73.0 ± 5.3%) (** *p* ≤ 0.01). (**B**) Examination of different sample collection methods revealed slightly lower methylation levels in surgical specimens (71.4 ± 6.5%) compared to endoscopic ones (72.1 ± 6.9%). Non-tumorous samples are represented in green, while tumorous specimens are marked in red. FF: fresh frozen samples, FFPE: formalin-fixed paraffin-embedded samples, E: endoscopic samples, S: surgical samples.

**Figure 3 ijms-23-11608-f003:**
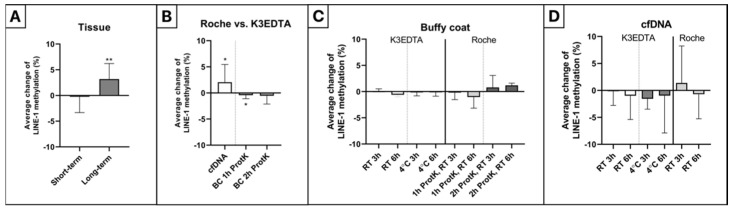
Impact of different storage conditions on LINE-1 methylation. (**A**) In the case of short-term storage, −0.9% methylation alteration was observed, while significantly elevated LINE-1 methylation values (+3.2%) were detected following long-term −20 °C storage of the isolated DNA (** *p* ≤ 0.01). (**B**) Significant LINE-1 methylation increase and decline were observed in cfDNA (+2.1%) and 1 h ProtK buffy coat (−0.4%) samples in comparison with standard K3EDTA tubes (* *p* ≤ 0.05), respectively. A non-significant, slight decrease was found in 2 h ProtK buffy coat specimens (−0.5%). (**C**) No significant LINE-1 methylation changes were detected in buffy coat samples depending on storage time or temperature in K3EDTA tubes (RT 3 h: +0.08%, 6 h: −0.6%; 4 °C 3 h: −0.2%, 6 h: −0.1%). In Roche collection tubes, non-significant methylation values with opposite changes were observed depending on the duration of ProtK digestion (1 h ProtK, RT 3 h: −0.2%, RT 6 h: −1%; 2 h ProtK, RT 3 h: +0.8%, RT 6 h: +1.2%). (**D**) In the case of whole blood samples taken into standard K3EDTA tubes, a non-significant decrease in cfDNA LINE-1 methylation could be noticed over time at both analyzed temperatures (RT 3 h: −0.1%, 6 h: −1%; 4 °C 3 h: −1.6%, 6 h: −1%). In cfDNA specimens isolated from Roche collection tubes, methylation alterations of +1.4% and −0.7% were found after 3 and 6 h of RT storage, respectively. cfDNA: cell-free DNA, RT: room temperature, 3 h: buffy coat and plasma separation after 3 h of storage, 6 h: buffy coat and plasma separation after 3 h of storage, 1 h ProtK: 1 h of ProtK digestion, 2 h ProtK: 2 h of ProtK digestion.

**Figure 4 ijms-23-11608-f004:**
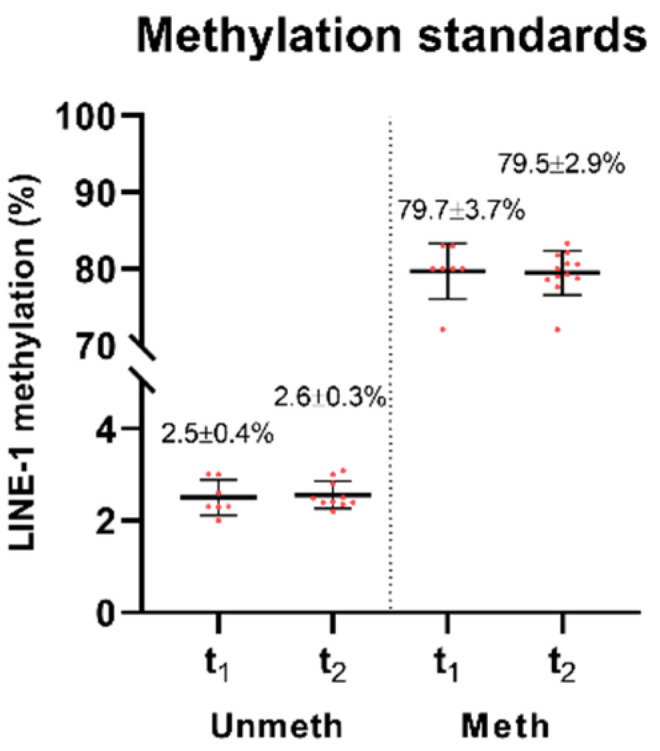
LINE-1 methylation of unmethylated and methylated bisulfite-converted DNA standards. No significant changes could be observed in both DNA controls over time. Unmeth: unmethylated standard samples, Meth: methylated standard samples, t1: first measurements, t2: measurements after two years.

**Figure 5 ijms-23-11608-f005:**
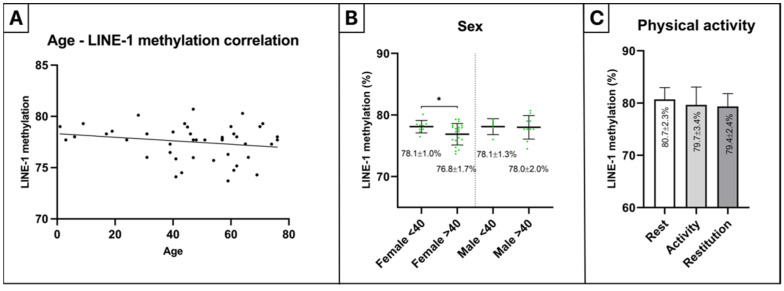
Analysis of LINE-1 methylation age and sex dependence in tissue, along with the impact of recent physical activity in plasma samples. No significant correlation was found between age and LINE-1 methylation (⍴ = −0.18, *p* = 0.25) (**A**), and no significant LINE-1 methylation alteration was noticed between sexes (**B**) in tissue samples. (**B**) Significantly lower methylation of tissue specimens was detected in female patients older than 40 years old (76.8 ± 1.7%) in comparison with the sample group of female patients under 40 years (78.1 ± 1.0%) (* *p* ≤ 0.05). (**C**) Decreasing LINE-1 methylation was observed at the highest heart rate (79.7 ± 3.4%) during the exercise and following the physical activity (79.4 ± 2.4%) compared to the initial resting phase (80.7 ± 2.3%) in plasma samples.

**Figure 6 ijms-23-11608-f006:**
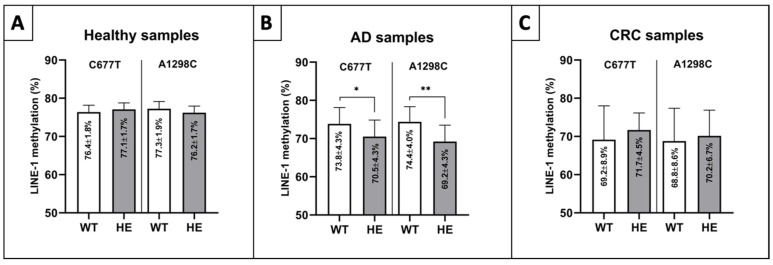
LINE-1 methylation alteration depending on MTHFR status in colorectal tissue samples. (**A**) No significant alterations could be observed in normal tissue samples. (**B**) Significantly lower LINE-1 methylation was detected in heterozygous AD samples of C677T and A1298C MTHFR mutations vs. WTs (* *p* ≤ 0.05, ** *p* ≤ 0.01). (**C**) A trend of elevation was found in the case of CRC LINE-1 methylation HE biopsies compared to WTs. WT: wild-type, HE: heterozygous alleles, AD: colorectal adenoma, CRC: colorectal carcinoma.

**Figure 7 ijms-23-11608-f007:**
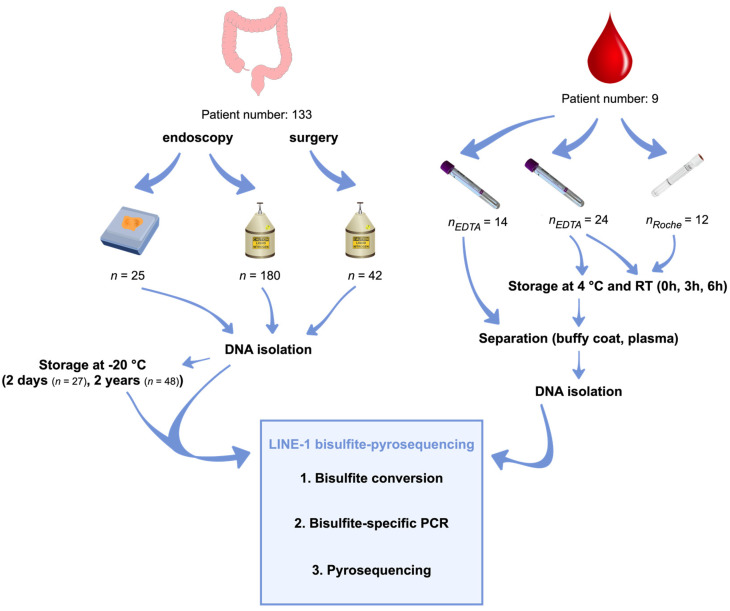
Summary of the sample collection and applied methods. Following the sample collection and fixation, DNA was isolated from colorectal N, AD, CRC, IBD along with paired AD-NAT and CRC-NAT tissue biopsies (patient number = 133). To investigate the impact of storage time, 27 specimens were analyzed after 2 days, and 48 samples were kept at −20 °C for 2 years before further examinations. Blood samples were collected into standard K3EDTA tubes to investigate the influence of physical activity on LINE-1 methylation results (patient number = 5). Furthermore, blood samples taken into conventional K3EDTA and Cell-Free DNA Collection Tubes were kept at RT and 4 °C for 0 h, 3 h, and 6 h to analyze the effect of different collection tubes, storage time, and temperature on LINE-1 methylation (patient number = 4). Buffy coat and plasma fractions were separated, and DNA specimens were isolated. Finally, DNA samples were bisulfite-converted followed by bisulfite-specific PCR of the LINE-1 promoter region, and the amplicons were pyrosequenced.

**Table 1 ijms-23-11608-t001:** Summary of the analyzed sample numbers.

		Tissue			Liquid Biopsy	
		FF		FFPE	Buffy Coat	Plasma
		Endoscopic Biopsy	Surgical Biopsy	Endoscopic Biopsy		
Healthy	Normal	45	-	9	19	4
	Athletes	-	-	-	-	5
Normal Adjacent Tissue to Colorectal Adenoma		23	-	-	-	-
Normal Adjacent Tissue to Colorectal Carcinoma		24	21	-	-	-
Colorectal Adenoma		37	-	12	10	-
Colorectal Carcinoma		36	21	4	10	-
Inflammatory Bowel Disease		15	-	-	-	-
Total		180	42	25	39	9

## Data Availability

Not applicable.

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
