# Peer review of "Methodological and Biological Factors Influencing Global DNA Methylation Results Measured by LINE-1 Pyrosequencing Assay in Colorectal Tissue and Liquid Biopsy Samples"

_ijms, 2022, doi:10.3390/ijms231911608_

Round 1

Reviewer 1 Report

Szigeti et al  used LINE1 methylation study as a reference to show that how sample collection, tissue storage, longevity of unprocessed samples affect clinical interpretation of DNA methylation study that may mislead to some context.

I have only concern of reproducibility of the data since authors used a single kit, I would recommend to be sure of this finding by using at least one among several available methods.

Author Response

Dear Reviewer 1,

We would like to thank you for your time and reviewing our manuscript entitled „Methodological and biological factors influencing global DNA methylation results measured by LINE-1 pyrosequencing assay in colorectal tissue and liquid biopsy samples” (Manuscript ID: ijms-1882158). Your valuable observation was taken into account and we hope that you will find our answer acceptable. 

Hereunder, we would like to respond to your specific comment.

Yours Sincerely,

Krisztina Szigeti, M. Sc.

Molecular Gastroenterology Laboratory

Department of Internal Medicine and Oncology

Semmelweis University

Korányi Sándor Str. 2/A

1083 Budapest, Hungary

Tel: +3620 825 0692

Email: szigeti.krisztina_andrea@semmelweis-univ.hu

General comments

According to the Reviewer’s observation, moderate editing of the English language is needed. We are grateful for drawing our attention to this important point, English editing was done by an English expert from Semmelweis University, dr. William J Kothalawala. The alterations are highlighted in red in the manuscript.

Specific comments

Comment: I have only concern of reproducibility of the data since authors used a single kit, I would recommend to be sure of this finding by using at least one among several available methods.

Response: Our original goal was to determine the global DNA methylation level. Following the overview of the recent literature, we have chosen the usage of ELISA assay specific to 5-methylcytosine and global DNA methylation estimation by LINE-1 pyrosequencing. However, ELISA is suitable for a rough estimation of DNA methylation changes, we considered it a favorable method for comparative analysis due to the fact that it is a quick and simple to perform, widely-used method. Furthermore, LINE-1 bisulfite-pyrosequencing has the lowest variability and the highest signal-to-noise ratio compared to several global DNA methylation analyzing methods. Besides, it is one of the most cost-effective, reliable techniques for global DNA methylation estimation, which results correlate well with the gold standard method, liquid chromatography combined with tandem mass spectrometry.

Firstly, we performed pilot measurements on colorectal healthy tissue and control DNA samples. According to these analyses, high deviation was detected using ELISA between technical replicants examined in the same plate (Table 1), along with experiments carried out in two years (Table 3), and also in two weeks (Table 2). Moreover, the alterations were not consequent. Taken together, we decided to conduct more examinations about the possible factors that may impact LINE-1 methylation levels, since the application of LINE-1 bisulfite-sequencing turned out to be more reliable in reproducibility in measuring the variance arising from bisulfite-conversion and PCR reactions (Table 4,5) and the influence of short-time storage (two days, involved in the manuscript). This analysis represented in Tables 4 and 5 was carried out according to the study of Irahara et al. (DOI: 10.2353/jmoldx.2010.090106), hence these data are part of our preliminary experiments, which are not involved in the main study. Please find below the ELISA results and the LINE-1 methylation values of technical replicates. Finally, we have an ongoing project about LINE-1 methylation estimation using whole genome sequencing by using Nanopore. This method analyzes unmodified (non-converted) DNA strands. Therefore the results represent higher values since all of the nucleotides and modified nucleotides can be distinguished. These data are planned to be involved in our next publication.

  1. Table Investigation of global DNA methylation level in the same biological colon tissue samples measured in duplicates in the same plate by 5mC ELISA assay.

Parallel samples measured on the same plate

Sample ID

Meth%

∆ Meth%

1

2

44

235.97

162.82

-72.90

45

173.88

181.11

7.23

46

82.04

94.40

12.35

  1. Table Investigation of the impact of short-time storage (two weeks) on global DNA methylation in control DNA samples measured by 5mC ELISA assay.

Short-time storage

Sample ID

Meth%

∆ Meth%

t1

t2

1

3.84

5.74

1.89

2

5.38

6.69

1.31

3

9.61

9.56

-0.05

4

23.01

16.31

-6.70

5

46.81

37.67

-9.13

6

84.44

80.94

-3.50

7

99.75

147.74

47.99

  1. Table Investigation of the impact of long-time storage (two years) on global DNA methylation in colon tissue samples measured by 5mC ELISA assay.

Long-time storage

Sample ID

Meth%

∆ Meth%

t1

t3

8

106.85

199.27

92.42

9

72.42

202.89

130.48

10

60.92

71.24

10.32

11

53.51

49.53

-3.97

12

50.51

44.23

-6.28

13

77.83

46.49

-31.33

14

24.57

31.58

7.01

15

80.10

102.79

22.69

16

37.86

23.24

-14.62

17

29.64

26.47

-3.17

18

29.64

18.53

-11.11

19

34.73

25.09

-9.63

20

55.87

26.03

-29.84

21

47.00

25.77

-21.23

22

41.88

80.86

38.97

23

38.41

40.69

2.27

24

54.28

77.71

23.42

25

31.85

78.77

46.92

26

73.47

64.55

-8.92

27

30.95

28.85

-2.10

28

34.73

28.59

-6.14

29

69.35

17.49

108.14

30

41.88

33.64

-8.24

31

148.84

116.77

-32.07

32

148.84

107.63

-41.21

33

26.03

27.32

1.29

34

30.50

31.01

0.51

35

58.34

52.42

-5.92

36

80.10

54.60

-25.50

37

69.35

35.68

-33.67

38

81.26

88.22

6.96

39

38.41

30.59

-7.82

40

32.78

21.50

-11.29

41

30.07

8.39

-21.68

42

28.38

18.02

-10.37

43

30.95

20.45

-10.49

  1. Table Investigation of global DNA methylation variaton in the same biological colon tissue samples arising from bisulfite conversion (BC) and PCR reactions using LINE-1 pyrosequencing. Three distinct bisulfite conversions were performed from the same DNA sample isolated from colon tissue followed by three separate PCR reactions of each bisulfite-converted specimen. Finally, the amplicons were pyrosequenced. BC: bisulfite-conversion

Sample ID

Reaction

Methylation %

Pos.1

Pos.2

Pos.3

Mean

47

BC 1

PCR 1

70.61

72.52

70.06

71.06

PCR 2

70.15

71.14

68.39

69.89

PCR 3

72.74

71.04

69.21

71

BC 2

PCR 1

73.53

70.55

68.21

70.76

PCR 2

73.41

71.41

70.08

71.63

PCR 3

74.94

72.2

69.22

72.12

BC 3

PCR 1

75.18

71.61

70.53

72.44

PCR 2

75.16

71.25

70.82

72.41

PCR 3

71.94

71.98

69.28

71.07

48

BC 1

PCR 1

78.89

74.42

74.35

75.89

PCR 2

77.54

74.01

72.86

74.8

PCR 3

77.16

74.35

72.11

74.54

BC 2

PCR 1

81.18

74.85

74.89

76.97

PCR 2

79.59

73.45

73.99

75.68

PCR 3

79.19

74.94

71.75

75.29

BC 3

PCR 1

80.98

74.76

72..21

75.98

PCR 2

80.28

74.03

73.6

75.97

PCR 3

81.29

74.67

74.89

76.95

19

BC 1

PCR 1

78.90

73.21

72.68

74.93

PCR 2

81.96

72.02

74.05

76.01

PCR 3

76.86

71.49

71.11

73.15

BC 2

PCR 1

75.99

70.56

71.75

72.77

PCR 2

76.61

72.05

71.19

73.28

PCR 3

77.98

73.01

72.78

74.59

BC 3

PCR 1

82

73.26

74.53

76.6

PCR 2

79.66

70.9

74.24

72.57

PCR 3

77.99

72.05

73

74.35

  1. Table Standard deviations between bisulfite-conversion and PCR reactions measured in the same biological colon tissue samples.

Sample ID

St. Dev between PCRs

St. Dev. between bisulfite-conversions

47

BC 1

0.66

PCR 1

0.90

BC 2

0.69

PCR 2

1.29

BC 3

0.78

PCR 3

0.63

48

BC 1

0.72

PCR 1

0.60

BC 2

0.88

PCR 2

0.61

BC 3

0.56

PCR 3

1.23

19

BC 1

1.44

PCR 1

1.92

BC 2

0.94

PCR 2

1.82

BC 3

2.02

PCR 3

0.77

Reviewer 2 Report

This paper summarizes interesting findings, but the overall structure and the way the discussion proceeds need improvement.

1)    As for the experimental method used, I would like to recommend that you add a schematic diagram as the first figure so that it can be understood even by unfamiliar readers.

2)    Various analysis conditions have been studied, but the description of the results has not been coherent, and it is difficult to see the relevance with the obtained knowledge. The analysis conditions and the finding on these conditions should be described first, and then what new knowledge has been obtained should be described.

3)     Factors have been found that affect the analysis results, and these data are useful for researchers who want to perform similar analyses, but on the other hand, there is no consensus on what conditions are best to analyze. Authors should give information on the most desirable conditions.

 I hope you will refer to these comments and improve your paper.

Author Response

Dear Reviewer 2,

We would like to thank you for reviewing our manuscript and drawing our attention to important topics that can improve our manuscript to a better scientific level. All of your valuable comments were taken into consideration and we modified the affected parts of the text accordingly. Please find our answers to your suggestions below; moreover, the altered sections are written in red in the manuscript. We hope you will find our work acceptable.

Yours Sincerely,

Krisztina Szigeti, M. Sc.

Molecular Gastroenterology Laboratory

Department of Internal Medicine and Oncology

Semmelweis University

Korányi Sándor Str. 2/A

1083 Budapest, Hungary

Tel: +3620 825 0692

Email: szigeti.krisztina_andrea@semmelweis-univ.hu

General comments

According to the Reviewer’s evaluation, the introduction section is needed to be improved. We endeavored to increase its quality by rearranging certain paragraphs, along with supplementing it with further information about DNA methylation estimating methods, and the potential clinical relevance of LINE-1 methylation.

“These above-mentioned biological aspects were chosen to be analyzed in relation to their influence on LINE-1 methylation results.”

“Enzyme-linked immunosorbent assay (ELISA) specific to 5-methylcytosine is a quick and simple, widely-used method for overall DNA methylation estimation, but it is able to measure rough differences in DNA methylation; moreover, it shows high inter- and intraplate variations [27].”

“Taken together, these literature data led us to prefer pyrosequencing in our previous works resulting in the realization of the importance to investigate methodological and biological circumstances that may impact the experimental results of LINE-1 pyrosequencing.”

“Methodological factors such as different manners of sample collection, and storage conditions can vary during an experimental or clinical investigation, which may have a significant impact on the methylation results. Since global DNA methylation analysis results can vary in a certain interval [31], but alterations lower than 5% in LINE-1 methylation can hold monitoring potential in post-operative CRC patients [8], standardized circumstances are needed for a possible diagnostic application of global DNA methylation estimation by LINE-1 pyrosequencing. The accuracy of this technique has already been investigated on plasma, colorectal FFPE tissue, and buffy coat samples [31,32,36]. However, a comprehensive analysis of the possible technical, experimental, and biological effects on LINE-1 methylation can provide useful information about the critical points of this procedure. Moreover, even though the application of unmethylated and methylated standards is not involved in the applied protocol, they may improve the comparability of the LINE-1 methylation values measured in different batches.”

“Firstly, we aimed to investigate the methylation values of distinct CpG sites in the LINE-1 promoter region analyzed by Pyromark Q24 CpG LINE-1 kit (Qiagen).”

Specific comments

Comment: As for the experimental method used, I would like to recommend that you add a schematic diagram as the first figure so that it can be understood even by unfamiliar readers.

Response: We accepted the helpful suggestion of the Reviewer regarding the illustration of the methods. We inserted it into the manuscript as “Figure 7”, and complemented the text.

The Material and methods section follows the Results according to the template of the journal, hence we were not able to include this figure as “Figure 1” but “Figure 7”. 

“The process of sample collection and the applied methods are presented in Figure 7.

Please find the figure in the attached PDF version of the answers.

“Figure 7. Summary of the sample collection and applied methods. Following the sample collection and fixation, DNA was isolated from colorectal N, AD, CRC, IBD along with paired AD NAT, and CRC NAT tissue biopsies (patient number=133). To investigate the impact of storage time, 27 specimens were analyzed after 2 days, and 48 samples were kept at -20°C for 2 years before further examinations. Blood samples were collected into standard K3EDTA tubes to investigate the influence of physical activity on LINE-1 methylation results (patient number=5). Furthermore, blood samples taken into conventional K3EDTA and Cell-free DNA Collection tubes were kept at RT and 4°C for 0h,3h, and 6h to analyze the effect of different collection tubes, storage time, and temperature on LINE-1 methylation (patient number=4). Buffy coat and plasma fractions were separated, and DNA specimens were isolated. Finally, DNA samples were bisulfite-converted followed by bisulfite-specific PCR of the LINE-1 promoter region, and the amplicons were pyrosequenced.”

Comment: Various analysis conditions have been studied, but the description of the results has not been coherent, and it is difficult to see the relevance with the obtained knowledge. The analysis conditions and the finding on these conditions should be described first, and then what new knowledge has been obtained should be described.

Response: We would like to thank the Reviewer for his/her valuable observation on how we could improve the clarity of the discussion section. The manuscript has been modified emphasizing firstly our goals in a certain paragraph, followed by our findings, and the discussion of the related recent literature.

“Our analysis regarding the effect of the chosen fixation method revealed significantly decreased methylation levels in FFPE compared to FF samples.”

“In the investigation of disparate sample collection methods, surgical and endoscopic samples were analyzed.”

“Firstly, we aimed to evaluate the LINE-1 methylation levels of DNA standards, along with DNA samples isolated from the same biological tissue biopsies and kept at -20°C for two days or two years. In the present study, short-term storage (two days) resulted in no significant LINE-1 methylation alteration in agreement with the study of Irahara et al. [31].”

„Our analyses reference to long-term storage of bisulfite converted, unmethylated, and methylated DNA standards revealed no significant impact on LINE-1 methylation level.”

„In the case of freshly collected whole blood samples, the impact of standard K3EDTA and blood collection tubes with stabilization reagents (Roche collection tubes) were investigated in buffy coat and plasma samples, along with the influence of different storage conditions. In the case of Roche collection tubes, significantly higher methylation levels were observed in cfDNA compared to standard K3EDTA tubes; moreover, significantly lower methylation values were found in buffy coat samples in the Roche vs. K3EDTA comparison analyzed by our research group. Accordingly, our previous study revealed significantly altered DNA methylation patterns of cfDNA samples obtained from CRC patients in stabilization blood collection tubes (Streck) compared to the conventional K3EDTA [49]. However, no significant methylation alteration was detected in white blood cells in the presence of DNA preservation agents published by Bulla et al. [50].”

„Distinct storage temperatures (RT, and  4°C), and disparate storage times (0, 3, and 6 hours) were examined as different storage variables in buffy coat and cfDNA specimens collected in conventional K3EDTA, along with Cell-Free DNA Collection Tube (Roche). No significant LINE-1 methylation alterations could be found either in buffy coat or cfDNA samples according to our investigations.”

„Moreover, storage time also occurred to be a considerable factor due to the significant global DNA methylation decline of white blood cells after 3 days of RT or 4°C  storage of blood samples based on the results observed by Huang et al. [55].”

„In contrast with our results about cfDNA LINE-1 methylation, 72 hours of storage of whole blood samples revealed slightly increased overall DNA methylation of cfDNA in EDTA tubes using reduced representation bisulfite sequencing according to a recent study [53].”

“In the second part of our study, we aimed to analyze the influence of distinct biological variables such as age, sex, physical training, and MTHFR status on the LINE-1 methylation pattern.”

“Regular physical training leads to an elevated LINE-1 methylation level [19,21]; hence, we involved the analysis of the influence of physical activity on LINE-1 methylation level, since, in a clinical application, it is highly important to know whether activities related to the daily routine could affect LINE-1 methylation results. A non-significant, minor LINE-1 methylation decrease was found during, and following recent physical activity compared to the resting phase in our investigations.”

“No remarkable alterations in healthy colorectal tissue were noticed in our investigation; meanwhile, higher LINE-1 methylation levels were detected in the case of samples with higher MTHFR activity in normal colonic mucosa according to Iacopetta et al. [57].”

Comment: Factors have been found that affect the analysis results, and these data are useful for researchers who want to perform similar analyses, but on the other hand, there is no consensus on what conditions are best to analyze. Authors should give information on the most desirable conditions.

Response: We thank the Reviewer for drawing our attention to this deficiency. We have supplemented the manuscript with a summary of our suggestions.

„In conclusion, methodological and biological variables can impact the LINE-1 methylation pattern measured by pyrosequencing on a relevant level, which should be taken into account in clinical and research fields as well.

In research, it is essential to collect and fix samples to be compared according to the same method; moreover, the experiments related to the same question are suggested to be conducted with kits purchased in the same production cycle. The importance of DNA standards applied from the beginning of the procedure is also highly emphasized, especially in a longitudinal examination.

In a medical investigation, standardization of the circumstances is crucial and more attention is needed, especially in the case of a multicenter sample collection. It is recommended to take the blood samples into the same type of collection tubes, store the samples according to the same protocol, and process them as soon as possible after the collection. Furthermore, the application of DNA standards is also suggested.

More investigations should be carried out concerning the effect of physical activity to determine an optimal resting period before the blood draw if it has been proven necessary. In regard to the other analyzed biological factors, such as age, sex, and MTHFR status are not relevant in the comparison of samples obtained from the same patient during a medical investigation; however, highly recommended to take them into account in experimental research.”

Round 2

Reviewer 2 Report

Thank you for your effort to improve the manuscript.  I now think that this manuscript has been improved and wii be give valuable information to wide variety of readers.  I think that this manuscript should be accepted.